# Therapeutic plasma exchange for coronavirus disease-2019 triggered cytokine release syndrome; a retrospective propensity matched control study

**Sultan Mehmood Kamran**[1]◉*, **Zill-e-Humayun Mirza**[1]◉, **Arshad Naseem**[2]◉, **Jahanzeb Liaqat**[3]◉, **Imran Fazal**[4], **Wasim Alamgir**[3,4], **Farrukh Saeed**[5], **Salman Saleem**[6], **Shazia Nisar**[6], **Muhammad Ali Yousaf**[3], **Asad Zaman Khan**[2], **Mehmood Hussain**[3], **Rizwan Azam**[1], **Maryam Hussain**[1], **Kumail Abbas Khan**[4], **Yousaf Jamal**[1], **Raheel Iftikhar**[7]◉

1 Department of Pulmonology, Pak Emirates Military Hospital (PEMH), Rawalpindi, Pakistan, 2 Department of Critical Care Medicine, Pak Emirates Military Hospital (PEMH), Rawalpindi, Pakistan, 3 Department of Neurology, Pak Emirates Military Hospital (PEMH), Rawalpindi, Pakistan, 4 Department of Medicine, Pak Emirates Military Hospital (PEMH), Rawalpindi, Pakistan, 5 Department of Gastroenterology, Pak Emirates Military Hospital (PEMH), Rawalpindi, Pakistan, 6 Department of Infectious Diseases, Pak Emirates Military Hospital (PEMH), Rawalpindi, Pakistan, 7 Department of Clinical Hematology, Armed Forces Bone Marrow Transplant (AFBMTC), Rawalpindi, Pakistan

◉ These authors contributed equally to this work.
* Sultanmajokoa79@hotmail.com

## Abstract

### Background

Cytokine release syndrome (CRS) plays a pivotal role in the pathophysiology and progression of Coronavirus disease-2019 (COVID-19). Therapeutic plasma exchange (TPE) by removing the pathogenic cytokines is hypothesized to dampen CRS.

### Objective

To evaluate the outcomes of the patients with COVID-19 having CRS being treated with TPE compared to controls on the standard of care.

### Methodology

Retrospective propensity score-matched analysis in a single centre from 1st April to 31st July 2020. We retrospectively analyzed data of 280 hospitalized patients developing CRS initially. PSM was used to minimize bias from non-randomized treatment assignment. Using PSM 1:1, 90 patients were selected and assigned to 2 equal groups. Forced matching was done for disease severity, routine standard care and advanced supportive care. Many other Co-variates were matched. Primary outcome was 28 days overall survival. Secondary outcomes were duration of hospitalization, CRS resolution time and timing of viral clearance on Polymerase chain reaction testing.

**Data Availability Statement:** All relevant anonymized data are within the paper and its Supporting Information files.

**Funding:** The author(s) received no specific funding for this work.

**Competing interests:** The authors have declared that no competing interests exist.

## Results

After PS-matching, the selected cohort had a median age of 60 years (range 32–73 in TPE, 37–75 in controls), p = 0.325 and all were males. Median symptoms duration was 7 days (range 3–22 days' TPE and 3–20 days controls), p = 0.266. Disease severity in both groups was 6 (6.6%) moderate, 40 (44.4%) severe and 44 (49%) critical. Overall, 28-day survival was significantly superior in the TPE group (91.1%), 95% CI 78.33–97.76; as compared to PS-matched controls (61.5%), 95% CI 51.29–78.76 (log rank 0.002), p<0.001. Median duration of hospitalization was significantly reduced in the TPE treated group (10 days vs 15 days) (p< 0.01). CRS resolution time was also significantly reduced in the TPE group (6 days vs. 12 days) (p< 0.001). In 71 patients who underwent TPE, the mortality was 0 (n = 43) if TPE was done within the first 12 days of illness while it was 17.9% (deaths 5, n = 28 who received it after 12$^{th}$ day (p = 0.0045).

## Conclusion

An earlier use of TPE was associated with improved overall survival, early CRS resolution and time to discharge compared to SOC for COVID-19 triggered CRS in this selected cohort of PS-matched male patients from one major hospital in Pakistan.

## Introduction

Globally, more than 40 million substantiated cases of coronavirus disease 2019 (COVID-19) have been recorded, with more than one million deaths [1]. Beyond supportive care, there are currently no proven effective treatment options for COVID-19 [2], although few treatment modalities such as remdesivir [3], tocilizumab [4], convalescent plasma (CP) [5] and mesenchymal stem cell (MSC) [6] therapy have shown early evidence of efficacy. With approximately 60% mortality [7] in critical cases, it is postulated that the fatal outcomes of COVID-19 are associated with excessive immune response. Cytokine release syndrome (CRS) manifests many abnormalities such as lymphopenia, high levels of C-reactive protein (CRP), high Ferritin, high D-dimers, high lactate dehydrogenase (LDH) and interleukin-6 (IL-6). These biochemical manifestations of CRS and significantly abnormal coagulation parameters were commonly found in severe and critically ill patients who did not survive [7]. Severe and critical COVID-19 patients are prone to develop sepsis, acute respiratory distress syndrome (ARDS) and/or multiple organ failure through immune dysregulation. It has been hypothesized that therapeutic plasma exchange (TPE), by removing pathogenic cytokines may have an additional role in managing early sepsis having onset less than 12 h [8]. Our study attempted to demonstrate that dampening of the cytokine syndrome (improvement of symptoms and settling of CRS markers) by using TPE when initiated within one week of CRS onset might be beneficial to the patients with COVID-19 having CRS. However, no prospective study on TPE has been conducted so far in patients with COVID-19 triggered CRS.

## Materials and methods

### Study design and setting

This study was a retrospective, propensity score matched (PSM) and single center, conducted at Pak Emirates Military Hospital (PEMH), Rawalpindi (Pakistan). PEMH is the largest

tertiary care military hospital of the country with a capacity of approximately 1200–1400 beds. The hospital is equipped with all necessary and advanced healthcare facilities for the management of COVID-19 including TPE, Extracorporeal membrane oxygenation (ECMO) and all investigational pharmacological modalities. Data of all hospitalized patients are maintained by COVID-19 Research and evaluation cell of the hospital. All data were fully anonymized before retrieval. Ethical review committee Pak emirates Military Hospital Rawalpindi approved the study. The data were extracted for patients with COVID-19 admitted with or developing CRS during their admission from April 1st to 31st July 2020. As the identity of the patients was not visible, informed written consent was waivered off by the president ethical review committee Pak Emirates Military Hospital Rawalpindi. Records of the patients were assessed during 3rd week of August 2020.

## Inclusion and exclusion criteria

Inclusion criteria included: (1) COVID-19 diagnosed by real time polymerase chain reaction (RT-PCR) positivity for severe acute respiratory syndrome corona virus 2 (SARS-CoV-2) (2) CRS at presentation or developing during hospitalization (3) age range 18–80 years and both genders (4) hospital admission (5) at least 1 completed session of plasma-exchange in patients included in TPE arm. Exclusion criteria were: (1) death within 48 hours of admission (2) severe septic shock at the time of admission (3) congestive cardiac failure (Ejection fraction <20%) (4) Those receiving immunotherapy, anti-thymocyte globulin or hematopoietic stem cell transplant in last 6 months (5) patients of hematological or solid organ malignancies (6) patients receiving other investigational drugs including tocilizumab, CP, remdesivir, or MSC therapy.

## Statistical analysis

Retrospective observational studies involving therapeutic interventions are often confounded by either measured or unmeasured baseline characteristics. As a result, baseline characteristics of treated subjects differ from untreated ones. To account for these systematic differences, we conducted a PSM analysis of patients of COVID-19 triggered CRS treated with or without TPE. PSM was performed on a cohort of patients meeting above inclusion and exclusion criteria. For the estimation of propensity score we used a logistic regression model (data matching Greedy) on NCCS statistical software v20.0.2 [9]. Data of 90 patients selected by this propensity-matching software are attached as S1 File. The number of controls were matched with the TPE treatment group in 1:1 matching. The distance calculation method used was the Mahalanobis distance including the propensity score, order for matching was random 1:1 and Caliper radius was set at 1*Sigma. For ensuring comparable groups, forced matching was done for disease severity, standard care and advanced treatment at disease escalation. Co-variate matching was done for age, duration of illness, symptoms at presentation, comorbidities, serum Ferritin, lactate dehydrogenase, d-dimers, C-reactive protein (CRP) levels, the absolute lymphocyte count (ALC), platelet count and oxygen requirement at the time of CRS diagnosis. Median and range were used for continuous variables while frequency and percentage were used to express categorical statistics. The chi-square test was used to evaluate differences in categorical variables while Students t-test or Mann-Whitney U test was used to evaluate continuous variables. Kaplan-Meier test was used for survival analysis and log rank was used to compare difference in the two groups. Cox-proportional hazards were used to generate hazard ratios (HRs) and 95% confidence intervals (CIs) for the outcome. P-value less than 0.05 was considered significant.

## Terms

**Confirmation of COVID-19.** COVID-19 was confirmed by positive real-time polymerase chain reaction (RT-PCR) on nasopharyngeal and/or oropharyngeal swabs done at Armed Forces Institute of Pathology (AFIP). The three specific genes of severe acute respiratory syndrome coronavirus 2 (SARS-CoV-2), namely the open reading frame (1a/b (ORF1a/b), nucleocapsid protein (N), and envelope protein (E) genes, were amplified by RT-PCR technology. Result was declared positive when ORF1a/b gene was positive, and/or N gene/E gene positive.

**The severity of disease.** It was defined according to the criteria designed by WHO [10]. Moderate disease was defined as COVID-19-positive case with lung infiltrates < 50% of total lung fields on X-ray chest/peripheral ground glass opacities (GGOs) on High-resolution computerized tomography (HRCT) chest scan but no evidence of hypoxemia. Severe disease was defined as COVID-19 pneumonia with evidence of hypoxemia [respiratory rate (RR more than 30/minute or partial pressure of oxygen (PaO2) on arterial blood gas (ABGs) less than 80 mmHg or PaO2/FiO2 (PF ratio) less than 300 or lung infiltrates more than 50% of the lung field]. Critical disease was defined if there was COVID-19 pneumonia with evidence of either respiratory failure (PaO2 less than 60 mmHg) or multiorgan dysfunction syndrome (MODS) measured by sequential organ failure assessment (SOFA) score more than 10 or septic shock (systolic BP less than 90 or less than 40mm Hg of baseline in hypertensive or urine output less than 0.5 ml/kg/hour). All categories of patients were managed in the high dependency unit of the hospital except those who required invasive ventilation during the course of illness.

**CRS.** It was defined by National guidelines for COVID-19 [11]. CRS was diagnosed as fever of equal to or more than 100˚F persisting for more than 48 hours in absence of documented bacterial infection and ANY of the following in the presence of moderate, severe or critical disease; (1) Ferritin more than 1000 mcg/L and rising in last 24 hours prior to CRS diagnosis (2) Ferritin more than 2000 mcg/L in patient requiring high flow oxygen or ventilation (3) Lymphopenia less than 800 lymphocytes/ul or lymphocyte percentage <20% and two of the following (a) Ferritin more than 700 mcg/mL and rising in the last 24 hours prior to CRS diagnosis (b) LDH more than 300 IU (reference 140–250 IU/L) and rising in the last 24 hours prior to CRS diagnosis (c) D-Dimer more than 1000 ng/mL (or more than 1mcg/ml) and rising in the last 24 hours prior to CRS diagnosis (d) CRP more than 70 mg/L and rising in the last 24 hours prior to CRS diagnosis, in absence of bacterial infection (e) If any 3 of above presents on admission no need to document rise.

**Standard of care (SOC).** It was as per institutional COVID-19 management guidelines. All patients with moderate, severe and critical COVID-19 received standard protocol of aspirin, famotidine, anticoagulation, vitamin C, vitamin D, oral zinc, awake Proning (if PaO2 < 80mmHg) and corticosteroids. All patients of CRS received methylprednisolone 1 mg/kg irrespective of disease severity.

**TPE procedure.** In addition to SOC, TPE was offered as a trial investigational therapy to willing patients with CRS. All patients were explained the investigational role of TPE in treatment of COVID-19. Written consent was taken from those who agreed for this treatment. TPE was performed once daily using COBE Spectra Apheresis machine version 7 (Manufacturer TERUMO BCT, Lakewood, CO, USA INC) with continuous flow centrifugation. Venous access was achieved using an ultrasound guided double lumen catheter (Arrow—12 FR) via the femoral vein. Patient's total blood volume was calculated as per Nadler's formula [12]. Anticoagulant acid dextrose ratio was 1:10 and flow rate 30–40 ml/minutes (Adjusted as per hemodynamic status). Patients' blood pressure, pulse and oxygen saturation were monitored throughout the procedure. The duration of procedure varied from 2–4 hours and 1.5 times total plasma volume was removed during each procedure. Replacement fluid was fresh frozen

plasma (FFP) and normal saline in 2:1 respectively. All procedures were performed in intensive care by Apheresis Department of PEMH and patients were shifted back to the high dependency unit immediately after procedure. TPE was continued once daily until recovery. The median follow-up time after last TPE session was 28 days. TPE related complications were also documented.

**Recovery.** It was defined by de-escalation of patients' condition from critical, severe and moderate disease to mild disease plus at least 2 of the following; serum Ferritin < 1000 ug/ml (and decreasing trend on two consecutive days), serum LDH normalization, C-reactive protein > 50% reduction (and decreasing trend in two consecutive days), Absolute lymphocyte count (ALC) > 1000.

**Study end points.** The primary end point was 28-day survival defined by discharge from hospital and remain symptom free on weekly follow-ups until 28th day after the last session of TPE. Patient discharge criteria included normalization of CRP, LDH, d-dimers and fall in serum Ferritin to less than 500 mcg/ml plus an afebrile period of at least 72 hours, and maintaining oxygen saturation more than 93% for at least 72 h without supplemental oxygen support. Secondary outcome measures were (1) duration of hospitalization (2) timing of PCR negativity (3) time to resolve CRS symptoms

## Results

Patient selection procedure is shown in Fig 1. On initial screening of data for the patients with COVID-19, 315 cases of CRS were found. After applying the exclusion criteria, 280 eligible patients were included in PSM analysis (data are available in zip folder attached as S1 File) Using 1:1 matching, 45 pairs of patients were formed, treated with or without TPE. The baseline characteristics of patients before and after matching are shown in Table 1.

### Before PSM

There were significant differences in TPE treated and untreated groups, which were addressed after matching. The overall eligible cohort was 245/280 (87.5%) males and 35/280 (12.5%) females, with a median age of 62 years (range 20–80 years) p-value 0.05, while 150/280

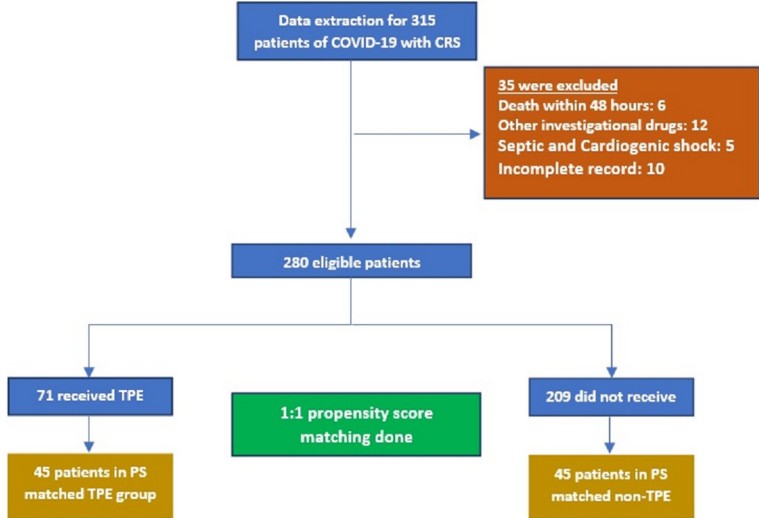

**Fig 1. Patient selection and matching flowchart.** COVID-19, coronavirus disease 2019; TPE, therapeutic plasma exchange; PS, propensity score.

**Table 1. Demographic and clinical characteristics of the study and control groups.**

| Characteristics | Before PSM | | | After PSM | | |
|---|---|---|---|---|---|---|
| | TPE (n = 71) | No TPE (n = 209) | P value | TPE (n = 45) | No TPE (n = 45) | P value |
| Age years, median (range) | 60 (32–80) | 64 (20–91) | 0.05 | 60 (32–73) | 60 (37–75) | 0.325 |
| **Age groups, n (%)** | | | <0.001 | | | 0.690 |
| 10–30 years | 0 | 4 (1.9%) | | - | - | |
| 30–50 years | 19 (26.7%) | 42 (20%) | | 8 (17.8%) | 7 (15.6%) | |
| 50–70 years | 45 (63.3%) | 105 (50.2%) | | 32 (71.1%) | 32 (71.1%) | |
| >70 years | 7 (10%) | 58 (27.7%) | | 5 (11.1%) | 6 (13.3%) | |
| **Gender, n (%)** | | | 0.235 | | | 0.306 |
| Male | 65 (92%) | 180 (86.1%) | | 45 (100%) | 45 (100%) | |
| Female | 6 (8%) | 29 (13.9%) | | 0 | 0 | |
| **Co morbidities, n(%)** | | | 0.137 | | | 0.98 |
| Obstructive air way disease | 3 (4.2%) | 10 (5.5%) | | 2 (4.4%) | 2 (4.4%) | |
| IHD | 5 (7%) | 11 (5.2%) | | 3 (6.7%) | 3 (6.6%) | |
| Down syndrome | 1 (1.4%) | - | | - | - | |
| DM | 5 (7%) | 23 (11%) | | 6 (13.3%) | 6 (13.3%) | |
| DM+HTN | 12 (17%) | 19 (9%) | | 2 (4.4%) | 2 (4.4%) | |
| IHD+DM+HTN | 6 (8.6%) | 17 (8.1%) | | 3 (6.7%) | 3 (6.7%) | |
| HTN | 7 (9.8%) | 24 (11.4%) | | 4 (8.88%) | 4 (8.88%) | |
| >3 co morbidities | 1 (1.4%) | 1 (0.47%) | | 4 (9%) | 4 (9%) | |
| RA+ILD | - | 1 (0.47%) | | - | - | |
| CVA | 1 (1.4%) | 1 (0.47%) | | - | - | |
| CKD | 1 (1.4%) | 1 (0.47%) | | - | - | |
| No | 29 (40.8%) | 100 (47.8%) | | 21 (46.7%) | 21 (46.7%) | |
| **Clinical features n(%)** | | | | | | |
| Cough | 59 (83%) | 168 (80%) | 0.61 | 36 (80%) | 37 (82.2%) | 0.787 |
| SOB | 58 (81%) | 144 (68.8%) | 0.04 | 36 (80%) | 40 (88.9%) | 0.244 |
| The duration of symptoms at admission, median (range) | 9 (1–25) | 5 (1–20) | <0.001 | 7 (3–22) | 7 (3–20) | 0.266 |
| **Laboratory values** | | | | | | |
| Absolute Lymphocyte count x $10^9$/l, median (range) | 754 (200–2100) | 800 (230–1960) | <0.001 | 700 (200–2100) | 790 (230–1400) | 0.692 |
| Platelet count x $10^9$/l, median (range) | 190 (70–1100) | 205 (56–450) | 0.448 | 180 (70–1100) | 187 (56–450) | 0.603 |
| CRP (ug/ml), median (range) | 120 (6–303) | 112 (56–390) | 0.268 | 145 (21–278) | 147 (56–260) | 0.284 |
| IL-6, median (range) | 95 (6–400) | 68 (7–679) | 0.206 | 78 (6–400) | 104 (7–178) | 0.116 |
| D-dimers, median (range) | 400 (150–1700) | 341 (200–1100) | 0.571 | 350 (150–1700) | 647 (300–1100) | 0.642 |
| Ferritin (ng/ml), median (range) | 1848 (378–16942) | 1505 (71–2832) | <0.001 | 1500 (336–7877) | 1410 (395–4500) | 0.218 |
| ALT (IU/l), median (range) | 99 (23–345) | 68 (21–288) | 0.057 | 64 (31–355) | 75 (28–288) | 0.744 |
| LDH (U/l), median (range) | 557 (235–1590) | 548 (273–1123) | <0.001 | 549 (235–1590) | 545 (273–1448) | 0.817 |
| **Cardiac biomarkers, n(%)** | | | 0.73 | | | 0.188 |
| Normal | 37 (52.1%) | 103 (49.2%) | | 23 (51%) | 28 (62.2%) | |
| Raised | 34 (47.9%) | 106 (50.8%) | | 22 (49%) | 17 (37.8%) | |
| **HRCT Chest findings, n(%)** | | | 0.09 | | | 0.37 |
| Typical | 65 (91.5%) | 164 (78.5%) | | 41 (91%) | 38 (84.4%) | |
| Indeterminate | 2 (2%) | 18 (8.6%) | | 1 (2.3%) | 1 (2.2%) | |
| Atypical | 3 (4.2%) | 23 (11%) | | 2 (4.4%) | 6 (13.4%) | |
| Normal | 1 (1.4%) | 4 (1.9%) | | 1 (2.3%) | - | |
| **Lung involvement (%), n(%)** | | | 0.07 | | | 0.607 |
| <50 | 17 (23.9%) | 84 (40.2%) | | 13 (28.8%) | 13 (28.8%) | |
| >50 | 53 (74.7%) | 121 (57.9%) | | 31 (69%) | 32 (71.2%) | |

*(Continued)*

**Table 1.** (Continued)

| Characteristics | Before PSM | | | After PSM | | |
|---|---|---|---|---|---|---|
| | TPE (n = 71) | No TPE (n = 209) | P value | TPE (n = 45) | No TPE (n = 45) | P value |
| Normal | 1(1.4%) | 4(1.9%) | | 1 (2.2%) | | |
| **Oxygen support Liter/min, (range)** | 10 (2–15) | 6 (0–15) | <0.001 | 10(4–15) | 10.5(0–12) | 0.730 |
| NIV | 34 (47.8) | 59 (28.2) | <0.001 | 19 (42.2%) | 19 (42.2%) | 0.66 |
| IV | 7 (10) | 6 (2.8) | 0.01 | 3 (6.6%) | 3 (6.6%) | 1 |
| **Disease severity,n(%)** | | | <0.001 | | | 1 |
| Moderate | 4(5.6%) | 71(34%) | | 3 (6.6%) | 3 (6.6%) | |
| Severe | 25(35.2%) | 110(52.6%) | | 20 (44.4%) | 20 (44.4%) | |
| Critical | 42(59.2%) | 28(13.4%) | | 22 (49%) | 22 (49%) | |
| **Other treatment,n(%)** | | | | | | |
| Steroids | 71(100%) | 209(100%) | - | 45(100%) | 45(100%) | 1 |
| Anticoagulation | 63 (88.7%) | 178 (85.1%) | 0.453 | 45(100%) | 45(100%) | 1 |
| **Outcomes** | | | | | | |
| Positive Day 7 PCR (67 and 189 evaluable) n(%) | 28 (41.7%) | 80 (42.2%) | 0.93 | 14(31%) | 15(33.3%) | 0.82 |
| Positive Day 14 PCR (32 and 152 evaluable),n(%) | 4 (12.5%) | 25 (19.6%) | 0.57 | 3(6.6) | 3(6.60 | 1 |
| Discharge days, median (range) | 10 (3–24) | 13(5–30) | 0.034 | 10(4–37) | 15(7–45) | 0.01 |
| Time for CRS resolution, median (range) | 7(3–20) | 11 (5–30) | 0.04 | 6(2–23) | 12(5–42) | 0.001 |

PSM; Propensity score matched, TPE; therapeutic plasma exchange, IHD; ischemic heart disease, DM; diabetes mellitus, HTN; hypertension, ALT; alanine transaminase, LDH; lactate dehydrogenase, Cardiac biomarkers; Creatine kinase-MB and troponins, HRCT; high resolution computerized tomography, NIV; noninvasive ventilation, IV; invasive ventilation, PCR; polymerase chain reaction, CRS; cytokine release syndrome.

(53.57%) had an age range between 50 and 70. Regarding chronic health conditions, 129/280 (46.07%) had no comorbidities. Hypertension (HTN) and diabetes mellitus (DM) either alone or along with other illnesses were the first and the second most common comorbidities respectively: (HTN; 85/280 (30.35%) and Type 2 DM; 82/280 (29.28%). HRCT chest scan showed the typical appearance in 229/280 (81.78%) with 174/280 (62.14%) having > 50% lung involvement. A total of 71 patients received TPE compared to 209 patients who did not receive TPE. A significant difference (p = 0.0045) was seen between early Plasma exchange as compared to late. Mortality in individuals who underwent Plasma exchange within the first 12 days of illness was 0 (n = 43). Mortality in those who underwent the procedure later on was 17.9% (deaths 5, n = 28).

Before PSM analysis, two groups differed significantly with respect to different demographic and clinical features as in Table 1.

## After PSM

After PSM, two groups (TPE versus non TPE) of 45 patients each had comparable characteristics. Regarding PSM-cohort (n = 90), 71.1% (32 TPE + 32 non-TPE) had age range between 50 and 70 (p = 0.690) with 46.7% (21 TPE + 21 non-TPE), (p = 0.98) having no comorbidities. After PSM, no female was left in either groups hence gender-based analysis could not be done. HRCT chest scan had more than 50% lung involvement in 63/90 (70%). Among 44/90 (48.8%) patients ventilated, 38/90 (42.2%) patients received Continuous positive airway pressure (CPAP) and 6/90 (6.6%) patients required mechanical ventilation. The cohort treated with TPE received a median of 2.25 sessions (range 1–5). Median time to start the first TPE session from the date of admission was 3.5 days (IQR; 2–5, mean;3.96) Only two patients developed TPE related complications (femoral artery puncture, Thrombophlebitis of femoral vein with

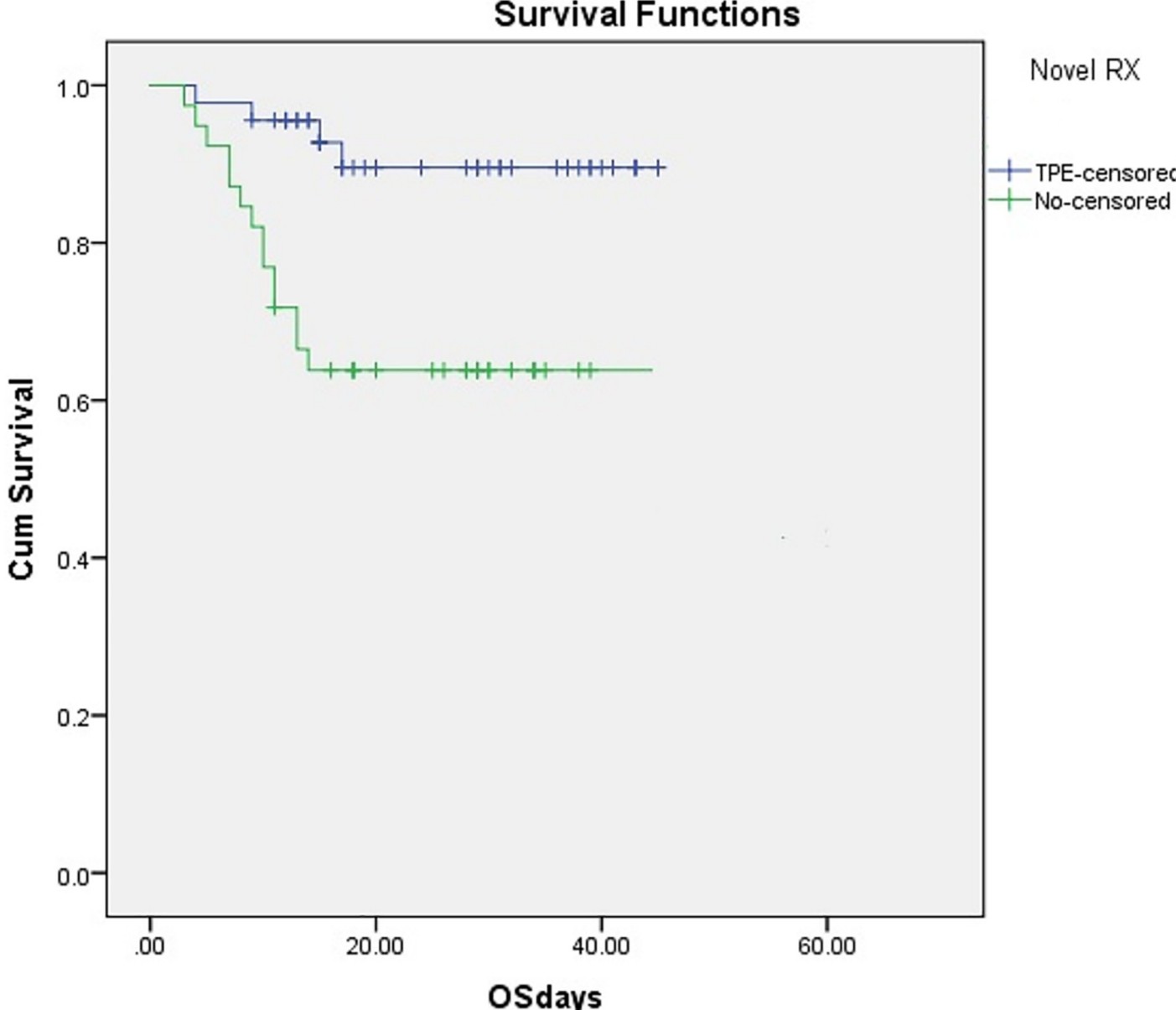

**Fig 2. Overall survival in TPE group (91.1%), 95% CI 78.33–97.76; as compared with the PSM controls (61.5%), 95% CI 51.29–78.76 (log rank 0.002), p<0.001.**
TPE, therapeutic plasma exchange, CI, confidence interval, PSM, propensity score matched, OS, overall survival.

DVT), which were managed optimally. Overall survival was significantly superior in the TPE group (91.1%), 95% CI 78.33–97.76; compared to PS-matched controls (61.5%), 95% CI 51.29–78.76 (log rank 0.002), p<0.001.(Fig 2). Cox regression analysis was performed to analyze the effect of covariate on survival outcome in both groups. After adjusting for age, comorbidities, disease severity and duration of symptoms, Overall survival (OS) in a TPE group remained superior to the PSM control group (p<0.001), HR 0.21 and 95% CI 0.07-.636.

Regarding effect of comorbidities in TPE group; (n = 45), overall survival (OS) was 100% for 21 patients without comorbidities compared to 83.4% for 24 patients with comorbidities. In patients not receiving TPE (n = 45), OS was 77% for 21 patients without comorbidities and

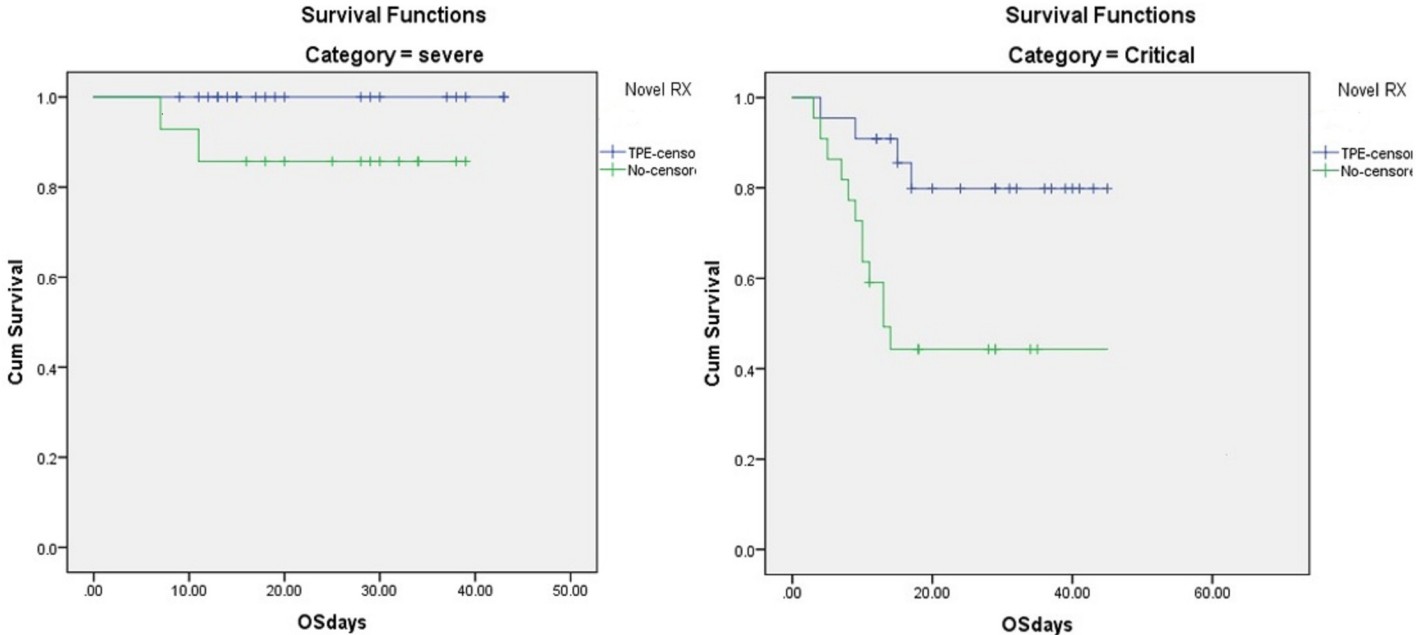

**Fig 3. Survival comparison in severe and critical COVID-19 between TPE and non-TPE cohorts.** (a) For severe COVID-19 patients, OS of 100% and 90% (p = 0.08) in TPE and non- TPE patients respectively (b) for critical COVID-19 patients, OS of 81.8% and 40.9% (p = 0.007) in TPE and non TPE patients respectively. COVID-19, Coronavirus disease 2019, TPE, Therapeutic Plasma exchange, OS, overall survival.

46% in 24 patients with comorbidities (p = 0.023). Out of 45 patients in each study group, 3 patients had moderate, 20 had severe and 22 had critical COVID-19 in each cohort respectively. Overall survival for patients with moderate, severe and critical COVID-19 was 100%, 100% and 81.8% for the TPE group compared to 100%, 90% and 40.9% for patients not receiving TPE (log rank 0.002) (Fig 3). The time of resolution of CRS was significantly reduced in the TPE group. From the time of admission until day 15, the cumulative incidence for normalization of CRS was 90% in the TPE group vs. 50% for PSM-controls. The Gray's test was applied to cater for completing risk and the difference was statistically significant (p < 0.001). Overall, 108/280 (38.6%) patients remained PCR-positive on day 7 of the first PCR positivity and 29/280 (10.35%) on day 14 of first PCR positivity with or without TPE. The median duration of hospitalization was significantly reduced in the TPE treated group compared to non-TPE controls (10 days vs. 15 days (p< 0.01).

## Discussion

This study demonstrates that addition of TPE to the SOC (inclusive of steroids) for moderate, severe and critical COVID-19 with CRS is associated with significant survival benefits especially in critical disease. TPE remarkably decreases the duration of hospitalization and resolution of CRS. However, PCR positivity on day 7 and 14 remained unchanged with addition of TPE. This finding is to be expected since evidence suggests the presence of non-replicable viral nucleic acid material only, after day 10 of onset of illness, being picked up by the PCR [13, 14]. To neutralize more comprehensively for biases associated with the selection of a particular mode of treatment, we stringently matched TPE and standard of care (SOC) treatment groups using a PSM analysis. The strength of the conclusion stems from the fact that many variables given in Table 1 in both arms had to be matched before analysis. TPE appears promising as an investigational therapy for several convincing reasons. First, TPE [15] has been used in

secondary hemophagocytic lymphohistiocytosis (sHLH) (recommendation III,2C; optimum role of TPE unestablished but low-quality evidence available), thrombotic microangiopathy secondary to various causes (various categories and strengths of evidence) and septic shock (recommendation III,2B; optimum role of TPE not established but moderate quality evidence available). As all or some of these pathologies may be present in severe to critical COVID-19 hence the role of TPE in treatment of COVID-19 may be partially justified. The cytokine profile of severe COVID-19 closely resembles sHLH [16] and it is also associated with venous and arterial thromboembolic complications [17] and septic shock [18]. Therefore, it was postulated that TPE will be similarly beneficial if used in COVID-19 triggered CRS. Second, TPE has also been successfully used previously for the managing severe infections such as 2009 H1N1 influenza A [19] and sepsis with multiorgan failure [8, 20, 21]. Third, TPE has been proposed as a possible supportive treatment of fulminant SARS-CoV-2 infection [22]. Moreover, TPE has also been shown to be effective in few case reports of COVID-19 [23, 24]. However, it has been argued that it's benefit in COVID-19 should be expected only in macrophage activation syndrome, or sepsis complicated with multiorgan dysfunction syndrome (MODS) [25]. It has been found that pro-inflammatory cytokines were significantly higher around 2nd week of illness [26] thus, key to success is early recognition of CRS, with early initiation of TPE. In this study, the median time to start the first TPE session from the date of admission was 3.5 days (IQR; 2–5, mean;3.96) and the patients who underwent TPE within 12 days of onset of symptoms had a remarkably improved survival. Hence, our study demonstrated better results of TPE when used earlier during the course of disease. The first reported study to our knowledge on the use of TPE in COVID-19 was conducted retrospectively on invasively ventilated patients receiving > 2 vasopressors which showed the greatest mortality benefit with TPE in these patients, (47.8% mortality in the TPE group vs. 81.3% mortality in the SOC group), (p = 0.05) [22]. In comparison, this study having larger sample size and done at various stages of illness (moderate, severe and critical cases), with only 6 patients on mechanical ventilation included, showed a significant benefit of TPE in severe and critical disease at a stage short of invasive ventilation. An emergency use of TPE in three patients with COVID-19 with acute respiratory distress syndrome was published as case series [24] that suggested that TPE had an immediate effect on the treatment of the CRS. Duration of hospitalization in these case reports was 18 to 25 days while it was 10 days in our study. This disparity might because of our discharge criteria that did not include PCR negativity which was one of the prerequisites in these reported cases. Even in the PSM control group, our study showed a mortality of 38.5% that is significantly lesser compared with a large retrospective study [7] showing a mortality of 60% in critical disease. The possible reasons might be the inclusion of moderate, severe and critical categories in our PSM cohort rather than only critical cases and the addition of steroids to all treatment groups having evidence of CRS.

It has been seen that coronaviruses such as severe acute respiratory syndrome (SARS)-CoV and Middle East respiratory syndrome-CoV (MERS) predominantly affect the male gender [27] and may be for the same genetic reasons, SARS-CoV-2 is also predominantly affecting male population as seen in pre PSM cohort of our study. Another reason for a predominant male population being infected in our study was our entitled clientele of the hospital was mainly from male gender. Nonetheless, this study had few limitations. First, it was a retrospective study which in itself has weaker evidence compared to prospective trials. Second, female gender was not represented in this study after PSM therefore results of this trial is exclusively applied to male gender only and a comparative analysis cannot be done. Third, although a strict PSM analysis was done, still all biases cannot be eliminated. Fourth, we did not follow up patients beyond 28 days from first TPE session due to lack of resources and huge influx of new COVID-19 cases. Lastly, in TPE procedure, we used centrifugation TPE machine rather than

continuous hemofiltration (CHF) which removes more IL-6 and similar cytokines molecular mass of 21 to 54.1 kDa [28] due to lack of availability of filters. Nevertheless, even after considering such limitations, using TPE, in addition to standard treatment in patients with COVID-19 may mitigate the cytokine storm. TPE shows promise, and we propose that large, multi-centric, randomized trials be designed to further investigate its role.

## Conclusion

In conclusion, TPE may be a lifesaving modality, with a statistically significant survival benefit, a decreased hospitalization time and an almost halved CRS resolution time, if started earlier at the onset of CRS in the treatment of severe and critical COVID-19 in the male population.

## Supporting information

**S1 File.**
(ZIP)

## Acknowledgments

I would like to express my deep gratitude to Professor Dr Zafar Ali Qureshi, Professor Dr Nadeem Ashraf Professor Dr Khalid raja and Professor Dr Malik Nadeem Azam for their patient guidance, enthusiastic encouragement and useful critiques of this research work. My grateful thanks are also extended to my colleagues; Dr Naveed Anjum, Dr Sohaib, Dr Faisal khan, Dr Samar, Dr marina and Dr Javeria for their help in collecting the data.

## Author Contributions

**Conceptualization:** Sultan Mehmood Kamran, Zill-e-Humayun Mirza, Arshad Naseem, Imran Fazal, Wasim Alamgir, Farrukh Saeed, Raheel Iftikhar.

**Data curation:** Sultan Mehmood Kamran, Arshad Naseem, Shazia Nisar, Mehmood Hussain, Rizwan Azam, Maryam Hussain, Kumail Abbas Khan, Raheel Iftikhar.

**Formal analysis:** Sultan Mehmood Kamran, Shazia Nisar, Raheel Iftikhar.

**Investigation:** Zill-e-Humayun Mirza, Arshad Naseem, Jahanzeb Liaqat, Wasim Alamgir, Farrukh Saeed, Salman Saleem, Muhammad Ali Yousaf, Asad Zaman Khan, Rizwan Azam, Maryam Hussain, Kumail Abbas Khan, Yousaf Jamal, Raheel Iftikhar.

**Methodology:** Sultan Mehmood Kamran, Zill-e-Humayun Mirza, Arshad Naseem, Jahanzeb Liaqat, Imran Fazal, Wasim Alamgir, Farrukh Saeed, Salman Saleem, Muhammad Ali Yousaf, Asad Zaman Khan, Rizwan Azam, Kumail Abbas Khan, Yousaf Jamal.

**Project administration:** Arshad Naseem, Imran Fazal, Wasim Alamgir, Farrukh Saeed, Salman Saleem.

**Resources:** Jahanzeb Liaqat, Imran Fazal, Wasim Alamgir, Farrukh Saeed, Shazia Nisar, Muhammad Ali Yousaf, Asad Zaman Khan, Yousaf Jamal.

**Software:** Muhammad Ali Yousaf, Mehmood Hussain, Raheel Iftikhar.

**Supervision:** Zill-e-Humayun Mirza, Arshad Naseem, Wasim Alamgir.

**Validation:** Zill-e-Humayun Mirza, Arshad Naseem, Imran Fazal, Salman Saleem, Asad Zaman Khan, Raheel Iftikhar.

**Visualization:** Salman Saleem, Asad Zaman Khan, Maryam Hussain.

**Writing – original draft:** Sultan Mehmood Kamran.

**Writing – review & editing:** Sultan Mehmood Kamran, Jahanzeb Liaqat, Shazia Nisar, Rizwan Azam, Maryam Hussain, Raheel Iftikhar.

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
