## [Decision Letter · Decision Letter 0]

12 Nov 2020

PONE-D-20-29085

Therapeutic plasma exchange for coronavirus disease-2019 triggered cytokine release storm; a retrospective propensity matched control study

PLOS ONE

Dear Dr. Kamran,

Thank you for submitting your manuscript to PLOS ONE. After careful consideration, we feel that it has merit but does not fully meet PLOS ONE’s publication criteria as it currently stands. Therefore, we invite you to submit a revised version of the manuscript that addresses the points raised during the review process.

Reviewers' suggest several major revisions. Please, amend your manuscript and resubmit.

We look forward to receiving your revised manuscript.

Kind regards,

Prof. Raffaele Serra, M.D., Ph.D

Academic Editor

PLOS ONE

Journal Requirements:

2. Thank you for including your ethics statement:  "Institutional Review board (IRB) approved the research via letter no A-28/EC/166/2020."   

 a.Please amend your current ethics statement to include the full name of the ethics committee/institutional review board(s) that approved your specific study.

 b.Once you have amended this/these statement(s) in the Methods section of the manuscript, please add the same text to the “Ethics Statement” field of the submission form (via “Edit Submission”).

For additional information about PLOS ONE ethical requirements for human subjects research, please refer to " ext-link-type="uri" xlink:type="simple">http://journals.plos.org/plosone/s/submission-guidelines#loc-human-subjects-research."

3. We noted that you refer to this study as an "interventional retrospective" cohort study. Since your study was retrospective, in order to avoid confusion we would suggest that you change the wording in your manuscript and avoid referring to this study as interventional.

4. We note that UNICEF is described as a study sponsor on your clinical trial registration. If any authors are affiliated to UNICEF, please state this on your manuscript cover page and in the online submission form. Additionally, if any funding was received from UNICEF, please state this in the Financial Disclosure field on the online submission form. Review the submission guidelines for detailed requirements.

5. Please ensure your Methods and reagents are be described in sufficient detail for another researcher to reproduce the experiments described. Specifically, please provide the source, product number and any lot numbers of the reagents purchased for your study.

6. Please include captions for your Supporting Information files at the end of your manuscript, and update any in-text citations to match accordingly. Please see our Supporting Information guidelines for more information: http://journals.plos.org/plosone/s/supporting-information

7. We note that your paper includes detailed descriptions of individual patients/participants; in the supporting information individuals are named.

As per the PLOS ONE policy (http://journals.plos.org/plosone/s/submission-guidelines#loc-human-subjects-research) on papers that include identifying, or potentially identifying, information, the individual(s) or parent(s)/guardian(s) must be informed of the terms of the PLOS open-access (CC-BY) license and provide specific permission for publication of these details under the terms of this license.

Please download the Consent Form for Publication in a PLOS Journal (http://journals.plos.org/plosone/s/file?id=8ce6/plos-consent-form-english.pdf).

The signed consent form should not be submitted with the manuscript, but should be securely filed in the individual's case notes.

Please amend the methods section and ethics statement of the manuscript to explicitly state that the patients/participants has provided consent for publication: “The individuals in this manuscript have given written informed consent (as outlined in PLOS consent form) to publish these case details”.

Additional Editor Comments:

The article is potentially interesting. Provided you revise your article according to reviewers' suggestions we will be happy to reconsider your work.

Reviewers' comments:

Reviewer's Responses to Questions

**Comments to the Author**

1. Is the manuscript technically sound, and do the data support the conclusions?

Reviewer #1: Yes

2. Has the statistical analysis been performed appropriately and rigorously? 

Reviewer #1: Yes

3. Have the authors made all data underlying the findings in their manuscript fully available?

Reviewer #1: Yes

4. Is the manuscript presented in an intelligible fashion and written in standard English?

Reviewer #1: Yes

5. Review Comments to the Author

Reviewer #1: The correct terminology in English for CRS is cytokine release syndrome. This should be corrected throughout the text.

This study looks at whether TPE may be a useful treatment option for COVID-19 disease. In order to appropriately segregate the patient population, the authors utilized as PSM study design. While this is an design, it ended up with a much reduced population of only 90 out of 280 patients. This reduced comparison led to some issues not addressed by the authors:

1) Primarily, it led to only a single female patient. This was not addressed by the authors other than that there were "primarily" males in the cohort (line 295-296). Due to the extreme differences between males and females, particularly in inflammation and inflammatory cytokine production, as well as susceptibility to more extreme forms of inflammatory disorders, this study should either be repeated using all of the 6 females with appropriate matches, or the single female should be excluded and the conclusions reported for male patients only.

2) The lymphocyte count in Table 1 is strange in that the difference in the count went from 45 without PSM to 90 with PSM even though the p-value increased to just outside of significance.

3) The IL-6 levels flipped following PSM restriction. Without the PSM, the TPE treated group had a higher median IL-6 level suggestive of more severe disease. With the PSM, the non-TPE treated group had a higher median IL-6 level, that was now nearing significance, suggestive of more severe disease. If the non-TPE group had more severe disease within each category of moderate, severe, critical, compared with the TPE group, the non-TPE group would be expected to do worse than the TPE group despite the TPE treatment.

The authors present better survival in the TPE treated group which is encouraging in Fig 2-3, though this is a bit confusing. There are 4 lines listed in the figure legend but only 2 on the graph. It doesn't look like the lines lie on top of each other, but if they do, you should offset the lines so that they can both be seen. If there are only 2 lines with censored data points then either remove the top 2 legends without the censor points, or change the symbol on the bottom 2 legends to just a + sign without the connecting line.

Figure 4 is incomplete and makes no sense. The label is just 1 and 2 and not TPE vs non-TPE. The axes are labeled normalizationCRS and cumulative normalizationCRS which are the same thing. I would think that this graph is supposed to show time to normalization of CRS. The data table also doesn't make since as it shows fewer patients at risk (either of CRS or normalization of CRS, I can't tell). If this is # of patients, then you should change the data table title.

The authors report that the study "demonstrates better results of TPE when used earlier during the course of infection" (line 275-276) but there is no data to show that. The survival curves demonstrate that TPE is better than not for less severe disease, but that is NOT the same thing as earlier in infection.

Also, the manuscript needs an overall editing to be consistent throughout in verbiage, abbreviations, etc.

6. PLOS authors have the option to publish the peer review history of their article (what does this mean?). If published, this will include your full peer review and any attached files.

Reviewer #1: No

---

## [Author Response · Author response to Decision Letter 0]

26 Nov 2020

Dated, 26th November, 2020

Dear 

Prof. Raffaele Serra, M.D., Ph.D

Academic Editor

PLOS ONE

Manuscript Number: PONE-D-20-29085

Therapeutic plasma exchange for coronavirus disease-2019 triggered cytokine release syndrome; a retrospective propensity matched control study

Thank you for your email. We are pleased to provide a revised version of our manuscript. We want to thank the reviewers for their time and effort in reviewing our manuscript. We sincerely appreciate their comments and have endeavored to respond with an acceptable revision.

Editor and Reviewer comments:

Third revision: (26/11/2020)

Thank you very much for pointing out the mistake. We have resubmitted the manuscript file after removing track changes.

Second revision: (25/11/2020)

Thank you very much for pointing out ambiguity in our ethical review statement. The full name of ethical review committee has been added. We have added following statement in the manuscript and in online submission section; Ethical review committee Pak emirates Military Hospital Rawalpindi approved the study. The data was extracted for the patients with COVID-19 admitted with or developing CRS during their admission from 1st April to 31st July 2020. As identity of the patients was not visible, the informed written consent was waivered off by the president ethical review committee Pak Emirates Military Hospital Rawalpindi. Records of the patients were assessed during 3rd week of August 2020

Earlier revision: (21/11/2020) 

 Answer: Thank a lot for guidance. All revised figures and manuscript are in accordance with styling of PLOS ONE

2. Thank you for including your ethics statement: "Institutional Review board (IRB) approved the research via letter no A-28/EC/166/2020." 

Answer:

Answer: Full name of Institutional Review Board is Ethical review committee Pak Emirates Military Hospital Rawalpindi, which has been added in ethics statement as well as in revised manuscript. b. Once you have amended this/these statement(s) in the Methods section of the manuscript, please add the same text to the “Ethics Statement” field of the submission form (via “Edit Submission”).

 Answer: The amendment has been added in Ethics Statement. 

3. We noted that you refer to this study as an "interventional retrospective" cohort study. Since your study was retrospective, in order to avoid confusion, we would suggest that you change the wording in your manuscript and avoid referring to this study as interventional.

 Answer: Thank you very much for pointing out. We have omitted the word “interventional “from the manuscript

4. We note that UNICEF is described as a study sponsor on your clinical trial registration. If any authors are affiliated to UNICEF, please state this on your manuscript cover page and in the online submission form. Additionally, if any funding was received from UNICEF, please state this in the Financial Disclosure field on the online submission form. Review the submission guidelines for detailed requirements.

 Answer: Thank you for pointing out. The statistician who helped in the research is working with UNICEF and her credentials were used to register the study. However, no funding was received for this study from any organisation. The name of UNICEF employee is Sumaira Irum (sirum@unicef.org) who was working with our research team on humanitarian basis during COVID pandemic. We did not include her name in author s list as she is non-medical professional. 

5. Please ensure your Methods and reagents are be described in sufficient detail for another researcher to reproduce the experiments described. Specifically, please provide the source, product number and any lot numbers of the reagents purchased for your study.

 Answer: TPE was performed by using COBE Spectra Apheresis machine version 7 (Manufacturer TERUMO BCT, Lakewood, CO, USA INC) having continuous flow centrifugation. This equipment is already available in our hospital and no new purchasing was done for the purpose of the study.

6. Please include captions for your Supporting Information files at the end of your manuscript, and update any in-text citations to match accordingly. Please see our Supporting Information guidelines for more information: http://journals.plos.org/plosone/s/supporting-information

 Answer: Thank you very much for pointing out. Now captions are added with supporting information files. 

7. We note that your paper includes detailed descriptions of individual patients/participants; in the supporting information individuals are named.

Answer: Regrettably, I failed to delete names of the patients from excel sheet. However, after your advice all the names are deleted. Omission is regretted.

As per the PLOS ONE policy (http://journals.plos.org/plosone/s/submission-guidelines#loc-human-subjects-research) on papers that include identifying, or potentially identifying, information, the individual(s) or parent(s)/guardian(s) must be informed of the terms of the PLOS open-access (CC-BY) license and provide specific permission for publication of these details under the terms of this license.

Please download the Consent Form for Publication in a PLOS Journal (http://journals.plos.org/plosone/s/file?id=8ce6/plos-consent-form-english.pdf).

The signed consent form should not be submitted with the manuscript, but should be securely filed in the individual's case notes.

Please amend the methods section and ethics statement of the manuscript to explicitly state that the patients/participants has provided consent for publication: “The individuals in this manuscript have given written informed consent (as outlined in PLOS consent form) to publish these case details”.

Answer: Thank you very much for this very important observation. As per advice of the reviewers, the statement of written informed consent has been added in the manuscript

Additional Editor Comments:

The article is potentially interesting. Provided you revise your article according to reviewers' suggestions we will be happy to reconsider your work.

Reviewer #1: The correct terminology in English for CRS is cytokine release syndrome. This should be corrected throughout the text.

Answer: The terminology has now been corrected as advised

This study looks at whether TPE may be a useful treatment option for COVID-19 disease. In order to appropriately segregate the patient population, the authors utilized as PSM study design. While this is an design, it ended up with a much reduced population of only 90 out of 280 patients. This reduced comparison led to some issues not addressed by the authors:

Answer:

1) Primarily, it led to only a single female patient. This was not addressed by the authors other than that there were "primarily" males in the cohort (line 295-296). Due to the extreme differences between males and females, particularly in inflammation and inflammatory cytokine production, as well as susceptibility to more extreme forms of inflammatory disorders, this study should either be repeated using all of the 6 females with appropriate matches, or the single female should be excluded and the conclusions reported for male patients only.

Answer: Thank you very much for pointing out a very important objection. We totally agreed that this study can not be applied to female gender because a very small sample size for female gender was left after PSM. Therefore, we have added this point in results, conclusion and in limitation of the study section.

2) The lymphocyte count in Table 1 is strange in that the difference in the count went from 45 without PSM to 90 with PSM even though the p-value increased to just outside of significance.

Answer: Thanks a lot for this important observation. The table shows median lymphocyte count with extreme values before PSM affecting p-value. After PSM although difference in lymphocyte count is 90 but the lymphocyte count of both groups has become more comparable due to propensity matching and hence p-value is not significant anymore.

3) The IL-6 levels flipped following PSM restriction. Without the PSM, the TPE treated group had a higher median IL-6 level suggestive of more severe disease. With the PSM, the non-TPE treated group had a higher median IL-6 level, that was now nearing significance, suggestive of more severe disease. If the non-TPE group had more severe disease within each category of moderate, severe, critical, compared with the TPE group, the non-TPE group would be expected to do worse than the TPE group despite the TPE treatment.

Answer: This observation of worthy reviewer is very important. PSM was done using NCCS software after incorporating patient’s data and matched pairs were than analyzed. Although IL-6 levels were higher in non-TPE group after PSM but difference was not significant statistically. (This point has been added in discussion as well.)

4. The authors present better survival in the TPE treated group which is encouraging in Fig 2-3, though this is a bit confusing. There are 4 lines listed in the figure legend but only 2 on the graph. It doesn't look like the lines lie on top of each other, but if they do, you should offset the lines so that they can both be seen. If there are only 2 lines with censored data points then either remove the top 2 legends without the censor points, or change the symbol on the bottom 2 legends to just a + sign without the connecting line.

Answer: Thanks a lot for highlighting this important point. Figures are corrected accordingly.

5. Figure 4 is incomplete and makes no sense. The label is just 1 and 2 and not TPE vs non-TPE. The axes are labeled normalization. CRS and cumulative normalization CRS which are the same thing. I would think that this graph is supposed to show time to normalization of CRS. The data table also doesn't make since as it shows fewer patients at risk (either of CRS or normalization of CRS, I can't tell). If this is # of patients, then you should change the data table title.

Answers: Thanks a lot for pointing point. Figure 4 has been removed and results written in text

6. The authors report that the study "demonstrates better results of TPE when used earlier during the course of infection" (line 275-276) but there is no data to show that. The survival curves demonstrate that TPE is better than not for less severe disease, but that is NOT the same thing as earlier in infection.

Answer: Thank you very much for pointing out this important observation. In our study the median time to start first TPE session from date of admission was 3.5 days which clearly indicates that CRS was picked up at the earliest and TPE started. A total of 71 individuals underwent plasma exchange in our study before PSM analysis. A significant difference (p=0.0045) was seen between early Plasma exchange as compared to late. Mortality in those individuals who underwent Plasma exchange within first 12 days of illness was 0 (n=43). Mortality in those who underwent the procedure later on was 17.9% (deaths 5, n=28). The same sentences have been added in the results section.

7. Also, the manuscript needs an overall editing to be consistent throughout in verbiage, abbreviations, etc.

Answer: Thank you very much for pointing out grammar/syntax related mistakes. I have tried to rectify as many as I could pick up and written as highlighted text. Nevertheless, if other mistakes are pointed out by respected editor, please send it back to us for rectification. We have updated reference no 1 also.

Thank you so much for these very astute observations and your kind consideration. Your suggestions have clearly improved our manuscript. Please do not hesitate to contact us should you have any further concerns or questions, and we would be more than happy to address them. Please allow us to resubmit updated manuscript for review and consideration for publication.

Sincerely,

Corresponding author for this manuscript is:

Dr. Sultan Mehmood Kamran

---

## [Decision Letter · Decision Letter 1]

14 Dec 2020

PONE-D-20-29085R1

Therapeutic plasma exchange for coronavirus disease-2019 triggered cytokine release syndrome; a retrospective propensity matched control study

PLOS ONE

Dear Dr. Kamran,

Thank you for submitting your manuscript to PLOS ONE. After careful consideration, we feel that it has merit but does not fully meet PLOS ONE’s publication criteria as it currently stands. Therefore, we invite you to submit a revised version of the manuscript that addresses the points raised during the review process.

The manuscript is potentially interesting, but some concerns should be addressed before we can reconsider for peer review again.

We look forward to receiving your revised manuscript.

Kind regards,

Prof. Raffaele Serra, M.D., Ph.D

Academic Editor

PLOS ONE

Additional Editor Comments (if provided):

The manuscript is potentially interesting but some concerns should be addressed.

Reviewers' comments:

Reviewer's Responses to Questions

**Comments to the Author**

1. If the authors have adequately addressed your comments raised in a previous round of review and you feel that this manuscript is now acceptable for publication, you may indicate that here to bypass the “Comments to the Author” section, enter your conflict of interest statement in the “Confidential to Editor” section, and submit your "Accept" recommendation.

Reviewer #1: (No Response)

2. Is the manuscript technically sound, and do the data support the conclusions?

Reviewer #1: Partly

3. Has the statistical analysis been performed appropriately and rigorously? 

Reviewer #1: Yes

4. Have the authors made all data underlying the findings in their manuscript fully available?

Reviewer #1: Yes

5. Is the manuscript presented in an intelligible fashion and written in standard English?

Reviewer #1: Yes

6. Review Comments to the Author

Reviewer #1: Due to the single female patient in the TBE-treated group, it is unacceptable to state that "female gender was less represented in this study...therefore results of this trial cannot be generalized on females and gender based comparative analysis cannot be done."

Those 4 patient need to be removed from the entire analyses. Since this is a retrospective study, this can be easily done by removing the data points for those individuals.

7. PLOS authors have the option to publish the peer review history of their article (what does this mean?). If published, this will include your full peer review and any attached files.

Reviewer #1: No

---

## [Author Response · Author response to Decision Letter 1]

14 Dec 2020

Answer: Thank you very much for giving us opportunity to revise our manuscript. As per your advice we have excluded four female patients from PSM cohort and replaced them with four similar male patients so that overall results not needed to be recalculated from the beginning. We took extreme care to select very similar 4 male patients having same age, duration of symptoms, comorbidities, HRCT chest score and pattern, levels of CRS markers, oxygen support, disease severity and TPE procedure. We have taken data of these 4 patients from our central registry and their identity have been masked. If this inclusion is not acceptable to you then please give us a chance to redo all statistics and Propensity score matching after excluding all female patients. Other than this, no observation has been raised by the respected editors/reviewers. We have also rectified all language/grammar related mistakes

---

## [Editor Report · Decision Letter 2]

18 Dec 2020

Therapeutic plasma exchange for coronavirus disease-2019 triggered cytokine release syndrome; a retrospective propensity matched control study

PONE-D-20-29085R2

Dear Dr. Kamran,

We’re pleased to inform you that your manuscript has been judged scientifically suitable for publication and will be formally accepted for publication once it meets all outstanding technical requirements.

Kind regards,

Prof. Raffaele Serra, M.D., Ph.D

Academic Editor

PLOS ONE

Additional Editor Comments (optional):

amended manuscript is acceptable
---

## [Editor Report · Acceptance letter]

28 Dec 2020

PONE-D-20-29085R2 

Therapeutic plasma exchange for coronavirus disease-2019 triggered cytokine release syndrome; a retrospective propensity matched control study 

Dear Dr. Kamran:

I'm pleased to inform you that your manuscript has been deemed suitable for publication in PLOS ONE. Congratulations! Your manuscript is now with our production department. 

Kind regards, 

on behalf of

Prof. Raffaele Serra 

Academic Editor

PLOS ONE